# Genome-scale reconstruction of Gcn4/ATF4 networks driving a growth program

Rajalakshmi Srinivasan[ID][1], Adhish S. Walvekar[ID][1], Zeenat Rashida[1], Aswin Seshasayee[ID][2]*, Sunil Laxman[ID][1]*

1 Institute for Stem Cell Science and Regenerative Medicine (inStem), GKVK post, Bangalore, India,
2 National Centre for Biological Sciences–TIFR, GKVK post, Bellary Road, Bangalore, India

* aswin@ncbs.res.in (AS); sunil@instem.res.in (SL)

**Data Availability Statement:** All genomic, transcriptome and ChIP raw data are available in NCBI-SRA under the accession PRJNA599001.

**Funding:** For this work, AS and SL were supported by grants from the DBT-Wellcome Trust India

## Abstract

Growth and starvation are considered opposite ends of a spectrum. To sustain growth, cells use coordinated gene expression programs and manage biomolecule supply in order to match the demands of metabolism and translation. Global growth programs complement increased ribosomal biogenesis with sufficient carbon metabolism, amino acid and nucleotide biosynthesis. How these resources are collectively managed is a fundamental question. The role of the Gcn4/ATF4 transcription factor has been best studied in contexts where cells encounter amino acid starvation. However, high Gcn4 activity has been observed in contexts of rapid cell proliferation, and the roles of Gcn4 in such growth contexts are unclear. Here, using a methionine-induced growth program in yeast, we show that Gcn4/ATF4 is the fulcrum that maintains metabolic supply in order to sustain translation outputs. By integrating matched transcriptome and ChIP-Seq analysis, we decipher genome-wide direct and indirect roles for Gcn4 in this growth program. Genes that enable metabolic precursor biosynthesis indispensably require Gcn4; contrastingly ribosomal genes are partly repressed by Gcn4. Gcn4 directly binds promoter-regions and transcribes a subset of metabolic genes, particularly driving lysine and arginine biosynthesis. Gcn4 also globally represses lysine and arginine enriched transcripts, which include genes encoding the translation machinery. The Gcn4 dependent lysine and arginine supply thereby maintains the synthesis of the translation machinery. This is required to maintain translation capacity. Gcn4 consequently enables metabolic-precursor supply to bolster protein synthesis, and drive a growth program. Thus, we illustrate how growth and starvation outcomes are both controlled using the same Gcn4 transcriptional outputs that function in distinct contexts.

## Author summary

Cell growth needs a ready supply of chemical building blocks (resources), which requires careful management. These resources provide energy, and help synthesize proteins through the process of translation. Some metabolites, such as methionine, push cells to grow even if resources are limited. How cells manage or obtain resources in these conditions is not well known. However, cells must activate 'master regulators', such as some

Alliance (grants IA/I/16/2/502711 and IA/I/14/2/501523). SL acknowledges funding from the Dept. of Biotechnology, Govt. of India (Grant number BT/PR13446/COE/34/30/2015) and intramural support. AS acknowledges funding from the Department of Atomic Energy, Govt. of India (project no. 12-R&D-TFR-5.04-0800). RS was supported by a SERB National Postdoctoral Fellowship, DST, Govt. of India (Grant number: PDF/2016/001877). The funders had no role in study design, data collection and analysis, decision to publish, or preparation of the manuscript.

**Competing interests:** The authors have declared that no competing interests exist.

transcription factors, to orchestrate programs to manufacture these building blocks. Here, using yeast cells as a model, we show how one such master regulator, the Gcn4/ATF4 transcription factor, can drive a growth program when activated by methionine. Although Gcn4 is well studied for its role in managing starvation, we find that it can perform similar functions when triggered by methionine, but in this case push cells to grow rapidly. It helps cells to produce a steady supply of amino acids (which are important building blocks), and simultaneously 'manage' the high rates of protein translation that are required in order to grow rapidly. This study highlights basic concepts of how cell growth is managed, and may be important for understanding similar observations from several cancers.

## Introduction

Understanding the organizational principles of transcriptional programs that define growth or starvation is of fundamental importance. In order to sustain growth, and thereby proliferation, cells must carefully manage available resources. In any growth program, the controlled supply of biosynthetic precursors is essential. These precursors include amino acids that drive protein translation, nucleotides (to make RNA and DNA), and several co-factors. Such a balanced cellular economy requires coordinated, genome-wide responses in order to manage metabolic resources and ensure coordinated growth outputs. Here, the model eukaryote, *Saccharomyces cerevisiae*, has been instrumental in building our general understanding of global nutrient-dependent responses, addressing how cells allocate resources, defining transcriptional and metabolic 'growth programs', as well as in uncovering general mechanisms of nutrient-sensing [1–9]. However, much remains unclear about how cells sustain the high requirement of biosynthetic precursors during growth programs.

Interestingly, studies from yeast and other systems show that some metabolites, even in nutrient-limited conditions, can induce cell growth programs (as observed at the level of transcription, signaling or metabolism) [10–12]. In this context, methionine (and its metabolite S-adenosyl methionine) turn on growth programs in cells [13–15]. In mammals, methionine availability correlates with tumor growth [16,17], and methionine restriction improves cancer therapy, by limiting one-carbon and nucleotide metabolism [18,19]. In yeast, supplementing methionine inhibits autophagy [20], activates growth master-regulators [13], and increases cell growth and proliferation [13,21]. At the level of global transcriptional and metabolic states, methionine triggers a hierarchically organized growth program. In this program, cells transcriptionally induce ribosomal transcripts, and key metabolic nodes including the pentose phosphate pathway, as well as all amino acid, and nucleotide biosynthesis pathways [15]. These are quintessential hallmarks of a cell growth program [22]. Therefore, using this controlled growth program, it may be possible to decipher universal regulatory features that determine a growth state. In particular, such a system can address how the metabolic program is coupled to the regulation of translation outputs. Unexpectedly, this previous study suggested that the transcription factor Gcn4 was critical for this growth program [15]. The role played by Gcn4 in this growth program was both unclear and unforeseen. This is in part because our current understanding of Gcn4 comes primarily from its role during starvation. Contrastingly, the role of Gcn4 during high cell growth is poorly explored.

Gcn4 (called ATF4 in mammals) is a transcriptional master-regulator, best studied for its roles during starvation and stress [23–26]. During amino acid starvation, the translation of Gcn4 mRNA increases, through the activation of the Gcn2 kinase, and subsequent eIF2-alpha

phosphorylation [23,27,28]. This resultant increase in Gcn4 protein allows it to function as a transcriptional activator, where it induces transcripts involved in amino acid biosynthesis. This allows cells to restore amino acid levels and survive starvation [23,26,29,30]. Most of the current knowledge of Gcn4 function comes from studying its roles during such amino acid and other nutrient starvations. Contrastingly, we found that in a growth program triggered by abundant methionine, cells induce Gcn4, in a context of high cell proliferation [15]. Other studies in several cancers also suggest that the mammalian ortholog of Gcn4, called ATF4, is critical to sustain high growth [31,32]. Since starvation and growth programs are considered to be opposite ends of a spectrum, we wondered what specific roles Gcn4 might execute during this growth program.

In this study, we find that Gcn4 controls essential metabolic components of a methionine-induced anabolic program, which allows the management of overall translation outputs. We elucidate the direct and indirect, methionine-dependent roles of Gcn4, and identify critical roles for this protein in controlling the metabolic components of this growth program, as well as in managing the induction of translation-related genes. During such a growth program, Gcn4 directly transcribes genes required for amino acids and transamination reactions, and indirectly regulates 'nitrogen' metabolic processes and nucleotide synthesis. Gcn4-enabled biosynthetic precursor supply is critical for appropriately maintaining a high translation capacity. Notably, comparing this function of Gcn4 during growth programs, to its known roles in starvation, reveals largely conserved transcriptional outputs of Gcn4 in both scenarios that lead to distinct outcomes for the cell (growth vs survival). Through this, we show how this transcriptional master-regulator uses its canonical outputs to enable a growth program, by ensuring specific amino acid supply in order to sustain sufficient translation capacity.

## Results

### Methionine induces a universal 'growth program'

Understanding the regulatory logic of transcriptional networks in growth programs is of fundamental importance. The role of the Gcn4 transcriptional master regulator has been well studied in contexts of nutrient starvation (as explained in the subsequent section), but is poorly studied during growth programs. Therefore, we first wanted to establish a relevant, universal system where the role of Gcn4 during a growth program could be rigorously studied. We utilized prior knowledge showing that methionine induces a growth program (at the transcriptional and metabolic level), while using lactate as a carbon source [15]. This includes the increased expression of ribosomal transcripts, and high metabolic flux through the pentose phosphate pathway, amino acid and nucleotide biosynthesis [15]. Notably, supplementing all non-sulphur amino acids together did not induce such a growth program [15]. Since the current studies were limited to utilizing lactate as a (non-conventional) carbon source, we first asked if this methionine response is universal, by studying the global transcriptional response to methionine supplementation in high glucose medium (the most preferred carbon source for yeast).

We performed a comprehensive gene-expression analysis and compared transcripts from cells growing in glucose (MM) or glucose supplemented with methionine (MM+Met). These analyses are shown in S1 Fig and S2 Fig, S1 Data, S2 Data, and described in extensive detail in the corresponding supplementary text (S1 Text). The methionine dependent transcriptional signatures of cells growing with glucose as a carbon source significantly overlap with the earlier studies where lactate was used as a carbon source [15]. Specifically, in order to compare the methionine dependent response across carbon sources, we first classified transcripts involved in amino acid biosynthesis and nucleotide biosynthesis genes as the 'anabolic gene group' (constituting 158 genes). Out of these 158 genes, ~40% of the genes are upregulated in

$MM_{gluc}$+Met (63 out of 158), with a $\geq$1.5 fold increase. 67% of these genes induced in $MM_{gluc}$+Met overlap with genes induced in $MM_{lactate}$+Met (Fisher exact test p-value $< 10^{-10}$). This indicates that the induction of transcripts representing anabolic genes by methionine is universal, regardless of the carbon source (S2D Fig). Similarly, we grouped ~ 750 transcripts representing 'translation related genes'. Out of 750 translation related transcripts, ~ 175 were upregulated in $MM_{gluc}$+Met. Similar to the anabolic genes considered earlier, the transcripts induced by methionine (with glucose as a carbon source) significantly overlapped with transcripts induced by methionine (with lactate as a carbon source) (Fisher exact test p-value $< 10^{-10}$) (S2D Fig and S2 Data). These results collectively indicate that the transcriptional response to methionine retains all the hallmarks of an anabolic growth program even when glucose is used as a carbon source. This includes the induction of appropriate metabolic genes (particularly all amino acid biosynthesis, nucleotide biosynthesis and transamination reaction related genes), as well as cytoplasmic translation related genes (S2 Fig, S2 Data, S3 Data). This induction of the translation machinery along with amino acid and nucleotide synthesis genes are classic signatures of an anabolic program [22,33,34]. These transcriptional changes functionally result in an appropriate metabolic state switch (increased *de novo* amino acid and nucleotide synthesis), as determined using a quantitative, targeted, stable-isotope pulsed LC/MS/MS based flux approach (S3 Fig). We therefore use this system (MM+Met) to address universal principles of cell growth regulation, and address the role of Gcn4 in this growth program.

## GCN4 is induced by methionine and controls a conserved transcriptional signature in both growth and starvation programs

Gcn4 (a transcriptional master-regulator), is best studied in contexts of starvation, and as part of the integrated stress response [23,24,26,35,36]. Many studies of Gcn4 function use pharmacological inhibitors of amino acid biosynthesis, such as 3-amino triazole (3-AT) or sulfo meturon (SM) to induce Gcn4, to address its role during starvation responses (where cell growth is minimal) [26,29,30,37,38]. In contrast, we had earlier observed that supplementing methionine induces Gcn4 [15], coincident with *increased* cell growth and proliferation. Since this is distinct from conditions of starvation and low growth, we wanted to understand what the role of Gcn4 was, during a growth program.

We addressed this question by using the methionine-induced growth program (as described in S2 Fig and S3 Fig, and the previous section). As a control, we first compared the growth of wild-type (WT) and *Δgcn4* cells in MM or MM+Met (Fig 1A). Consistent with the data described in the previous section, the WT cells had higher growth when supplemented with methionine, and severely reduced growth (in methionine) upon the loss of Gcn4 (Fig 1, S4 Fig). We next asked if Gcn4 protein is induced in methionine-supplemented glucose medium. Indeed, Gcn4 protein levels substantially increase when methionine is supplemented (MM+Met) (Fig 1B and S5A Fig). This observation reiterates that Gcn4 can be induced by growth signals (methionine) irrespective of carbon source. We therefore dissected how much of this anabolic program is mediated by Gcn4. To address this, we compared the transcriptomes of WT and *Δgcn4* cells in MM+Met (S5B Fig), and found a clear Gcn4-dependent global response in the presence of methionine. ~900 transcripts were differentially expressed in *Δgcn4*, compared to WT cells in MM+Met. Here, 514 transcripts were upregulated, and 398 transcripts were downregulated in *Δgcn4* cells in the presence of methionine (fold change cutoff of > = 2 fold) (S5B Fig and S1 Data). As a control, in only MM medium (without supplemented methionine), far fewer transcripts (~160) showed any differential expression at all in *Δgcn4* relative to WT (S5B Fig). These data show that Gcn4 has a critical role for the methionine-dependent growth program in glucose.

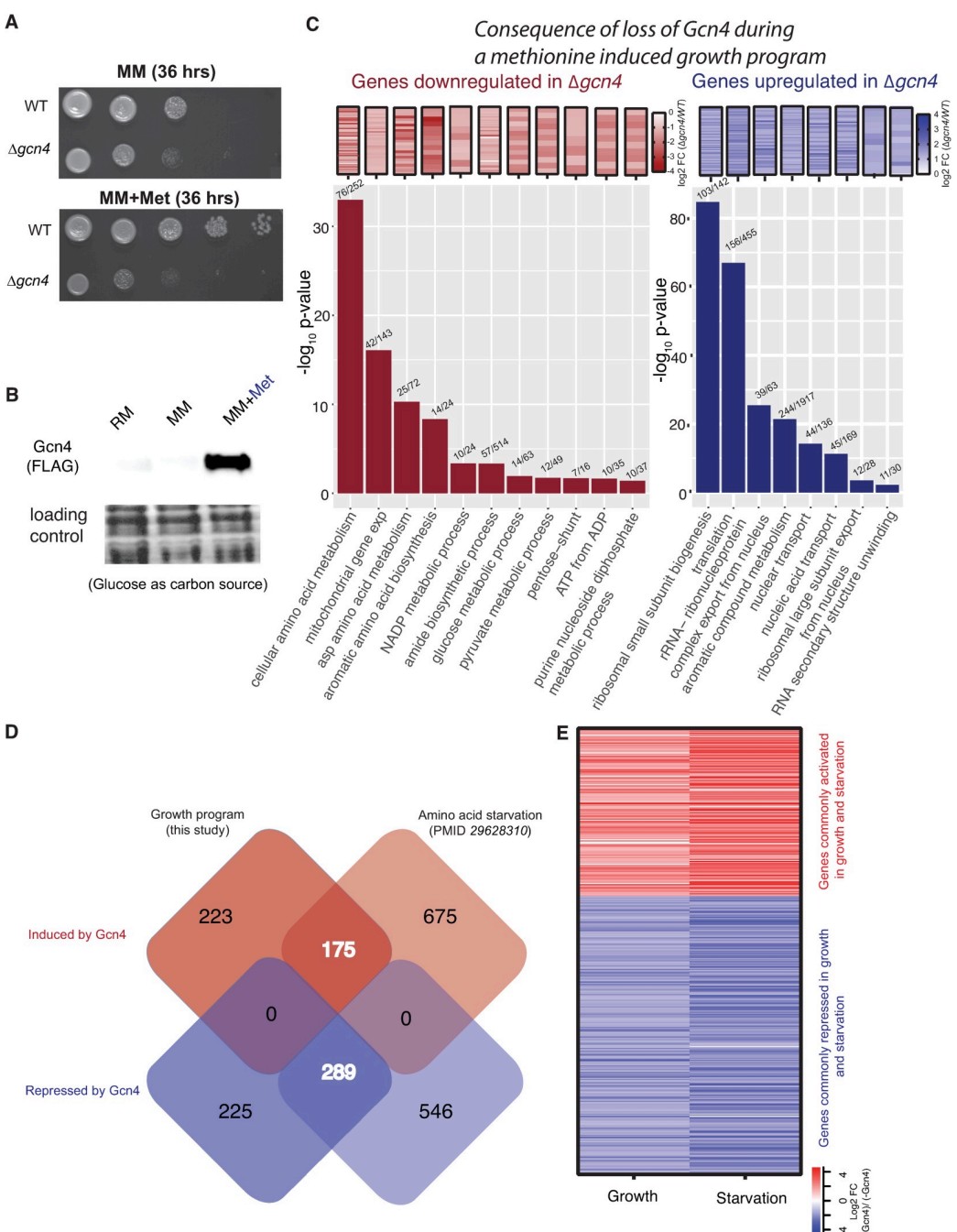

**Fig 1. GCN4 is strongly induced by methionine and controls a conserved transcriptional outcome in both growth and starvation programs.** A. Serial dilution based growth assay, comparing WT and *Δgcn4* cells growing in the presence or absence of methionine. WT cells show better growth when supplemented with methionine. Deletion of Gcn4 lead to severely reduced growth in the presence of methionine. Also see S4 Fig. B. Gcn4 is induced by methionine addition. A representative western blot shows high Gcn4 protein levels in MM+Met (Gcn4 tagged with the FLAG epitope at the endogenous locus). RM—rich medium, MM—minimal medium without amino acids and with glucose as a carbon source, and MM+Met—minimal medium without amino acids and with glucose as a carbon source supplemented with 2mM methionine. Also see S5A Fig. C. GO based analysis and grouping of transcripts down regulated in *Δgcn4* (red color scheme) and GO based analysis and grouping of the transcripts up-regulated in *Δgcn4* (blue color scheme). The bar graphs show the GO term plotted against the $\log_{10}$ p-value. The actual changes in expression of the relevant genes in each category are shown in the heat maps above each bar. The numbers of genes found in each GO group are indicated above each bar. All the terms shown here are significantly enriched terms with the corrected p-value < 0.05 (hypergeometric test, Bonferroni Correction). Also see S3 Data and S5B Fig for gene expression volcano plots. D. The Venn diagram shows the

number of differentially expressed genes that overlap, from data obtained from distinct cell growth conditions where Gcn4 levels are high. The boxes on the left are data from this study (methionine induced growth program), while the boxes on the right use data from a severe amino acid starvation condition (sulfo meturon addition) [41]. Also see S6 Fig and S6 Data. E. A heatmap showing fold changes in transcript amounts, for the overlapping genes common to growth and starvation conditions (as shown in Fig 1D).

To understand the global consequences of the loss of Gcn4 during this growth program, we used a GO-based analysis to categorize the most altered groups of transcripts. Upregulated transcripts in *Δgcn4* show a significant enrichment for 'cytoplasmic translation', 'ncRNA processing', 'RNA maturation' and 'RNA methylation' (Fig 1C). This surprisingly showed that the transcripts associated with protein translation, which were already induced by methionine, further increased in the absence of Gcn4. i.e. Gcn4 partially represses cytoplasmic translation even in a growth program. In contrast, transcripts that are downregulated in *Δgcn4* cells are primarily involved in amino acid biosynthetic processes, nucleotide biosynthetic processes, mitochondrial translation, NADP metabolic processes and pyruvate metabolism (Fig 1C and S3 Data). Collectively, these data show that Gcn4 is essential for the induction of genes involved in metabolic processes (which comprise the majority of the methionine induced anabolic program), but partially represses translation.

Here, we note an unusual observation relevant to growth programs. In studies of responses to starvation, the induction of Gcn4 represses ribosomal genes, and induces amino acid biosynthesis genes [30,39,40]. In contrast, in this methionine-induced growth program, ribosomal and translation-related transcripts are induced even as Gcn4 is also induced. The loss of Gcn4 further increases ribosomal transcripts. This is consistent with the possibility that Gcn4 appropriately keeps the extent of ribosomal transcript induction in check, while the ribosomal transcript induction occurs through independent regulation. Therefore, despite growth and starvation being at opposite ends of a spectrum, the core role of Gcn4 in either state might be conserved. To further address this possible role of Gcn4, we compared the overlap of Gcn4 dependent, induced or repressed transcripts in this growth program, with existing data from a starvation program where Gcn4 has high activity. This dataset comes from a study where Gcn4 was strongly induced, by inhibiting amino acid availability using a chemical inhibitor of amino acid biosynthesis (sulfometuron or SM) [41]. Notably, we find that 44% of the transcripts activated by Gcn4 and 56% of the genes repressed by Gcn4 in the methionine dependent growth program overlap with the transcripts activated and repressed in the SM dependent starvation condition (Fisher exact test, $p<10^{-10}$) (Fig 1D and 1E). We independently carried out a similar analysis using data from a distinct study where Gcn4 expression was increased using an inducible promoter (no chemicals were used to induce starvation) [42]. This comparison also showed a significant overlap between the transcripts regulated by Gcn4 under different modes of induction (S6 Fig). A GO grouping of the genes which overlap between the growth and the starvation condition suggests a well conserved role of Gcn4 in inducing amino acid biosynthetic genes, and in repressing translation related genes (S6 Data).

Two points emerge from these analyses. First, the role of Gcn4 seems to be conserved regardless of whether cells are in a growth or starvation program. This conserved role appears to be to increase transcripts related to amino acid and nucleotide biosynthesis (which are all required for anabolism), while repressing translation related transcripts. However, during a growth program, there is already an induction of translation related transcripts (as seen in Fig 1). Therefore, in this context, Gcn4 tempers the extent of induction of translation related transcripts, while during starvation Gcn4 represses ribosomal gene transcription below that of non-starved cells.

## Gcn4 binds to its target gene promoters related to metabolism during a growth program

Which components of the transcriptional outputs in this growth program does Gcn4 directly regulate, and how does this compare to the known, direct roles of Gcn4 during starvation? To address this, we performed chromatin immunoprecipitation (ChIP)- sequencing of Gcn4 in MM and MM+Met conditions. Notably, this uniquely integrates directly comparable information from the Gcn4 ChIP-seq, with a matched global transcriptome, *during a growth program*.

First, we asked what the direct Gcn4 binding targets are, when Gcn4 is induced by methionine. We performed ChIP of Gcn4 (with a FLAG-epitope incorporated into the C-terminus in the endogenous *GCN4* locus), using cells grown in MM and MM+Met. The MM condition essentially functions as a control. We considered peaks that are represented in both the biological replicates for further analysis, using very well correlated biological replicates (S5C Fig). Here, we identified 320 Gcn4 binding peaks in the cells grown in MM+Met, whereas, there were no consensus peaks observed in replicate samples of cells grown in MM (Fig 2A and S4 Data). The enhanced Gcn4 occupancy on the target gene promoter in MM+Met condition was further validated using ChIP-qPCR analysis (S8 Fig). This shows that the GCN4 occupancy on DNA increases in the presence of methionine.

Next, we analyzed the Gcn4 binding signals around the transcription and translation start site of the genes found within 750bp around the identified peaks. Transcription start site (TSS) data are available for cells growing in rich, glucose medium (the nearest possible condition to that used in this study) in the YeasTSS database [43]. The TSS classified using the CAGE method from this database was used for our analysis. A majority of the Gcn4 binding peaks in MM+Met are found upstream of these annotated transcription start sites (Fig 2B). A similar analysis with the translation start sites of the target genes shows higher read coverage upstream of the translation start site (S7 Fig). We further analyzed the genomic features of the identified peaks using the HOMER program [44]. We observed a clear enrichment of Gcn4 binding to the promoter region of the targets. 263 out of 320 peaks are found within the promoter region of target genes (-1kb to +100bp around the TSS), while the remaining peaks bind at intergenic regions (11), exons (17) or close to transcription termination sites (29) (Fig 2C and S4 Data). This shows that during a growth program, Gcn4 binding is primarily restricted to promoter sites of target genes.

We next searched for the enrichment of sequence motifs in the peaks identified in the MM +Met condition using the MEME-suite [45]. We found that these peaks were enriched for the conserved Gcn4 binding motifs found previously under amino acid starvation conditions [30,46,47]. About 81% (260 out of 320) of the peaks that we identify have at least one of the variants of the Gcn4 binding motif 'TGANTCA' (Fig 2D), showing that Gcn4 in this context still primarily recognizes its high-affinity DNA binding motif.

Finally, how does this compare to studies of Gcn4 activity during starvation, particularly when amino acid biosynthesis is inhibited using pharmacological agents [29,30]? A previous study of Gcn4 function during amino acid starvation indicated substantial Gcn4 binding to the regions within ORFs of genes, as well as to promoter regions [29]. To compare this study from a starvation program with our data from a growth program, for the non-coding and the ORF peak regions reported in the previous study [29], we calculated the Gcn4 binding signal in our Gcn4 ChIP seq data (from the MM+Met condition). Notably, we find that the signal in ORF peaks is significantly lower than the non-coding peak under MM+Met condition (p-value $< 10^{-8}$), whereas a similar analysis performed using the Gcn4 ChIP-seq data from [29] show little differences in the signal intensity between ORF and Non-coding peaks (p-value of 0.002) (Fig 2E). As a contrasting comparison, we used a dataset from a distinct nutrient

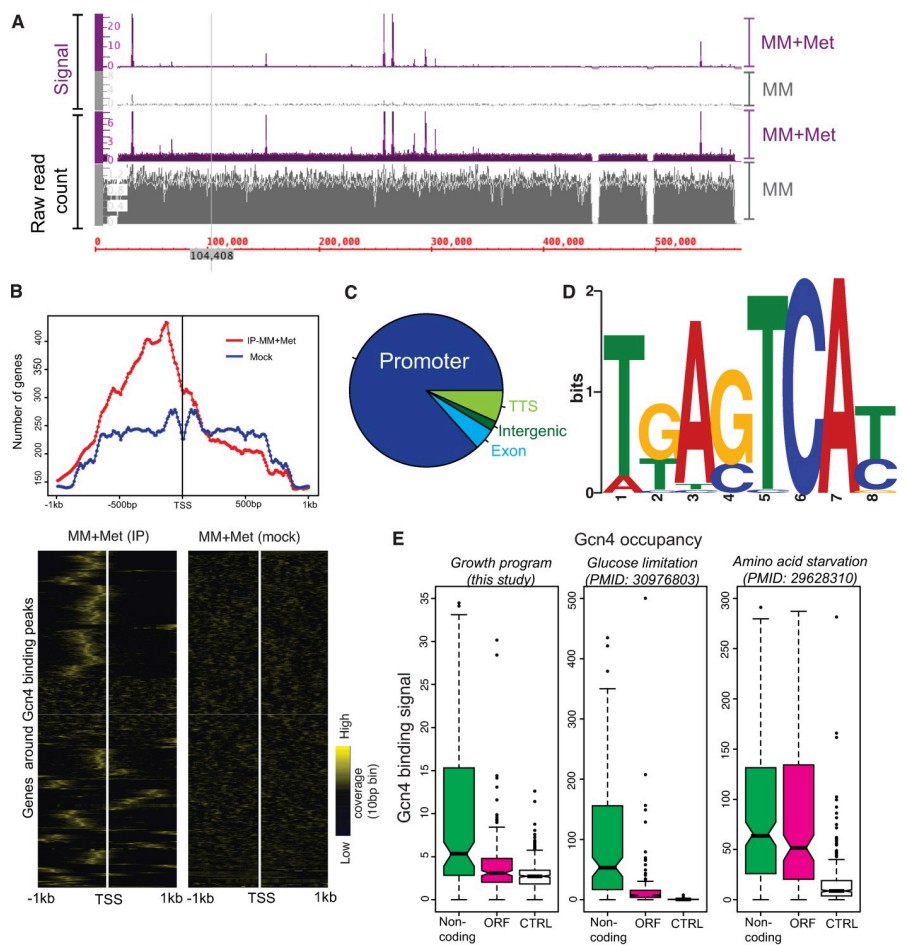

**Fig 2. Gcn4 binds to its target gene promoters related to metabolism during a growth program.** A. Genomic tracts showing Gcn4 binding to DNA regions in MM and MM+Met. Raw read counts and signal around the binding region of Gcn4 are shown. B. Top) Density plots showing that most target genes have Gcn4 binding peaks upstream of the Transcription Start Site (TSS) in the ChIP samples (red), whereas no such enrichment of peaks is observed in mock samples (blue). Here, all the genes that fall in the vicinity of 750bp around the identified Gcn4 binding peaks are considered as target genes. (Bottom) A heat map showing read coverage for Gcn4 binding, including 1kb upstream and downstream of predicted/known transcription start sites (TSS) of target genes. The heat map on the left shows read coverage in IP samples, and on the right shows coverage in mock-IP (control) both from cells grown in MM+Met. Read coverage around 1kb upstream and downstream of the TSS of the target genes were calculated separately for the genes coded on the positive strand and the negative strands, and merged. The heat map was generated without any clustering of the columns. The 'computeMatrix' and 'plotProfile' functions of the deepTools package were used for generating read coverage matrix. Also see S7 Fig, which shows read coverage for Gcn4 in the context of the translation start site (ATG) of each gene. C. A pie chart showing the genomic features of the identified peaks annotated using 'annotatepeak' function of the Homer tool [44]. D. Consensus binding motifs identified in Gcn4 binding peaks from MM+Met conditions. E. Boxplots, showing the Gcn4 binding signal corresponding to different genomic features, under distinct growth scenarios. For Gcn4 binding in non-coding and open reading frame (ORF) regions (as reported in a previous study [29]), we compared the Gcn4 binding signal in the Gcn4 ChIP sequencing data from cells in MM +Met (current study), or under different starvation conditions (amino acid starvation [29], or glucose limitation [48]). Notably, in either MM+Met or during glucose limitation the Gcn4 binding signal in ORF peaks is significantly lower than the non-coding region peaks ($p<10^{-8}$). Contrarily, under amino acid starvation [29], the Gcn4 binding signal found in ORF and non-coding regions are similar.

environment [48], where glucose was limited in a chemostat, but without any limitation in amino acids or nitrogen sources. In this dataset, the regions of DNA occupancy of Gcn4 was similar to that observed during our growth program, with a majority of Gcn4 occupancy at promoter regions of target genes (Fig 2E). These analyses show that during a growth program,

or homeostatic adjustments to altering nutrients during growth, the direct targets of Gcn4 remain specific, and restricted to the promoter regions of genes. This contrasts with conditions where amino acid biosynthesis is severely restricted, and where Gcn4 occupancy may extend to ORF regions as well. Collectively, our data shows that regardless of the mode of Gcn4 induction, and whether cells are in a growth or starvation program, it binds specifically to a highly conserved motif. Thus, the global role of Gcn4 during either a growth program, or in a starvation response appears well conserved.

## Direct and Indirect targets of Gcn4 during a growth program

We next asked how much of the Gcn4-dependent transcriptional response is directly regulated by Gcn4, and what its specific targets are. To identify direct targets of Gcn4 (in this methionine-dependent context), we overlaid the transcriptome data (*Δgcn4 vs WT* in MM+Met) with the ChIP-seq data (from MM+Met). Out of the 398 transcripts that are downregulated in *Δgcn4*, 133 are direct targets of Gcn4 (S4 Data). Contrastingly, Gcn4 directly regulates only 24 out of 514 upregulated transcripts (S9A Fig, S9B Fig, and S4 Data). These results strengthen the role of Gcn4 as a transcriptional activator. GO-based analyses of directly activated targets of Gcn4 reveal a significant enrichment of amino acid biosynthetic genes (Fig 3A). The indirectly activated targets are enriched for nucleotide biosynthesis, the pentose phosphate pathway, and mitochondrial translation (Fig 3A, S3 Data). In addition to the amino acid biosynthetic genes, Gcn4 directly activates genes involved in other critical functions, particularly the Sno1 and Snz1 genes (pyridoxal synthase), required for transamination reactions that lead to amino acid synthesis, and Nde1- the NADH dehydrogenase (S9C Fig). In contrast to the Snz1 and Sno1 pair that is bidirectionally activated by Gcn4, the Trm1 and Mdh2 pair of genes are bidirectionally repressed by Gcn4 (S9D Fig). These data show that in the presence of methionine, Gcn4 directly increases the expression of primarily the amino acid biosynthetic arm, whereas the methionine-dependent activation of nucleotide biosynthetic genes, pentose phosphate pathway, mitochondrial translation related genes are indirectly regulated by Gcn4. Collectively, the metabolic component of the methionine-dependent growth program is directly regulated by Gcn4.

As presented earlier, in the presence of methionine Gcn4 directly upregulates transcripts encoding multiple amino acid biosynthetic enzymes (Fig 3A). In this context, subsets of amino acid biosynthetic pathways are induced. Interestingly, every single gene of arginine biosynthetic pathway, and nearly every gene of lysine, histidine and branched chain amino acid biosynthetic pathways are directly activated by Gcn4 (Fig 3B and 3C, S9E Fig). This suggests that Gcn4 might be critical for the supply of particularly arginine and lysine, during the methionine mediated anabolic program.

We therefore estimated the functional contribution of Gcn4 towards individual amino acid biosynthesis in this methionine-dependent context, particularly that of arginine and lysine. For this we used a targeted LC/MS/MS based approach to measure amino acid synthetic flux, based on stable-isotope incorporation after a pulse of $N^{15}$ labeled ammonium sulfate (also see S1 Text and S7 Data for more details). Consistent with the transcriptome data, we found an induction of amino acid biosynthesis in MM+Met, compared to MM. The biosynthesis of arginine and lysine was strongly induced (Fig 3D). Expectedly, the loss of *GCN4* severely decreased the flux towards amino acid biosynthesis, and resulted in a near-complete loss of arginine and lysine biosynthesis (Fig 3D, S12 Fig, S7 Data). This reiterates that Gcn4 has a vital role in increasing the amino acid pools required during a methionine induced growth program, particularly regulating the synthesis of arginine and lysine.

How does the role of Gcn4 during this growth program compare to its role during amino acid starvation? To understand this, we analyzed a publicly available ChIP seq data of Gcn4,

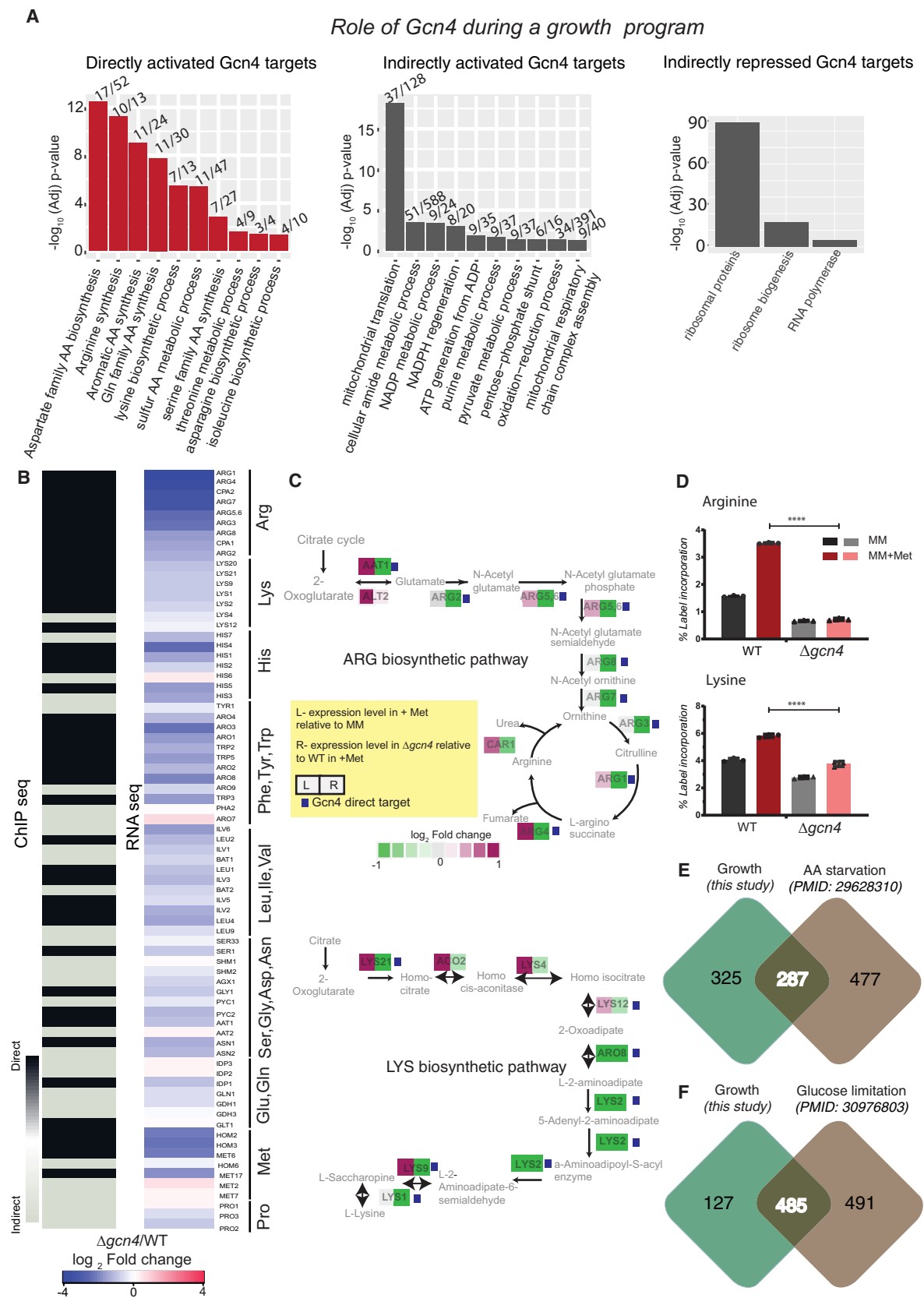

**Fig 3. Direct and Indirect targets of Gcn4 during a growth program.** A. Role of Gcn4 during a growth program (methionine addition): Bar plots shows enriched GO terms and the corresponding -log$_{10}$(p-value) for the genes which are directly or indirectly activated, or directly/indirectly repressed by Gcn4 when methionine is supplemented (growth program). The numbers of genes found in each GO group are indicated above each bar. Also see S9B Fig. B. Comparing direct targets of Gcn4 regulon, and gene expression profiles of WT and *Δgcn4* cells (Gcn4 dependence) in a growth program. The heat map on the left shows whether the indicated gene (involved in amino acid biosynthesis) is directly or indirectly regulated by Gcn4 based on ChIP-Seq data from cells in MM+Met medium. The black color indicates a direct target of Gcn4 and grey indicates an indirect target (Gcn4 does not bind the promoter of this gene). The heat map on the right shows the gene expression fold changes in *Δgcn4* relative to WT cells, grown in MM+Met medium. C. (Top panel) Representative pathway maps of the arginine biosynthetic pathway. This map shows the fold change in gene expression due to methionine (MM+Met compared to MM) in WT cells (left box), and the change in gene expression due to loss of Gcn4 (WT compared to *Δgcn4*) in the presence of methionine (MM +Met) (right box). Genes that are direct targets of Gcn4 are also indicated with a small purple box next to the gene name. (Lower panel) A representative pathway map of the lysine biosynthetic pathway, represented similar to that of arginine biosynthesis. D. Increased arginine and lysine biosynthesis in a methionine dependent growth program depend entirely on Gcn4. Data from quantitative LC/MS/MS based metabolic flux analysis experiments, using N$^{15}$ ammonium sulfate labeling to estimate new amino acid synthesis (based on label incorporation) in a methionine and Gcn4 dependent manner, are shown. The comparisons are between WT and *Δgcn4* cells treated identically in MM+Met medium. The data are presented for arginine, lysine. Also see S12 Fig. *p<0.05,**<0.01 (t-test). Also see S3 Table for MS parameters. E. Overlap between potential Gcn4 binding targets identified by ChIP-seq in a growth program (this study), vs the targets identified under a severe amino acid starvation response [29]. Correlation between the Gcn4 binding signals between these conditions are shown in the S11 Fig. F. Overlap between potential Gcn4 binding targets identified by ChIP-seq in a growth program (this study), vs the targets identified during glucose limitation [48]. Also see S11 Fig.

where Gcn4 was induced by amino acid starvation (due to SM treatment) (34). First, we compared potential Gcn4 targets, which are present 750bp around the Gcn4 peaks, identified in both the growth and the starvation condition. A 47% overlap was observed between the Gcn4 targets during the methionine induced growth program, and under amino acid starvation [29] (Fig 3E and S6 Data) (Fisher's Exact test P < 10$^{-10}$). We also compared the targets identified in our study with a distinct, starvation regime, where cells were limited for glucose [48]. About 80% of the targets identified in this study overlap with that of the Gcn4 targets identified in the glucose limitation study [48] (Fig 3F and S6 Data) (Fisher's Exact test P < 10$^{-10}$). We note a positive correlation of Gcn4 binding signals between growth and starvation conditions (S11 Fig). Specifically, for direct and indirect targets activated by Gcn4, the expression correlation between growth and the starvation conditions were compared. While the expression of both direct and indirect targets show positive correlation between growth and starvation conditions, direct targets show a higher correlation (Pearson correlation r = 0.43) between growth and starvation conditions, compared to the indirect targets (r = 0.3) (S10 Fig). These data indicate that the Gcn4 targets, particularly the activation of amino acid biosynthetic genes, are well conserved irrespective of the growth status of the cell.

Finally, in agreement with the previous ChIP-seq studies in starvation conditions [29,30], we find that Gcn4 indirectly represses translation related genes, except for the following-RPL14B, RPS9A, RPL36B and RRP5, which are directly repressed by Gcn4 in this condition (S9A Fig and S9B Fig). The distinction is that when methionine is present, ribosomal genes are induced, but Gcn4 appears to temper the extent of this induction (as the loss of Gcn4 in this condition further increases ribosomal genes). Therefore, through this repressive activity, Gcn4 might enable cells to manage the extent of ribosomal gene induction due to methionine.

To summarize, the role of Gcn4 in a methionine-dependent growth program can be broken into two parts. First, Gcn4 directly induces amino acid biosynthesis genes, as well as transamination reactions. Gcn4 is absolutely critical for the high rates of synthesis of amino acids (particularly arginine and lysine), nucleotide biosynthesis genes and the PPP, which together contributes to the methionine induced anabolic program. The ribosomal/translation related genes that are induced by methionine in WT cells are further induced upon the loss of Gcn4 in this condition, suggesting that Gcn4 indirectly tempers the extent of ribosomal gene induction due to methionine. Notably, the core function of Gcn4, which is to increase amino acid (and

nucleotide) synthesis, remains unchanged when cells are in a growth state or dealing with starvation.

## Gcn4 globally represses arginine/lysine-enriched genes, including the translational machinery

From the data presented thus far, it is clear that Gcn4 helps supply cells with several metabolites, particularly the amino acids arginine and lysine, when methionine triggers a growth program. Given this critical function of Gcn4 in arginine and lysine supply, we wondered if there were correlations between the presence of arginine and lysine encoding codons within transcripts of specific categories, and the global gene expression programs controlled by Gcn4. Although the amino acid compositions of proteins are evolutionarily optimized, our understanding of amino acid supply vs demand remains inadequate [49,50]. As amino acids are the building blocks of proteins, translation naturally depends on available intracellular amino acid pools. We therefore asked if there were categories of proteins that were particularly enriched for arginine and lysine, and whether this had any correlation with Gcn4 function. For this, we divided the total number of proteins in the *S. cerevisiae* genome into three bins based on the percentage of arginine and lysine content of the protein (%R+K). The bin1 comprises 1491 proteins with the lowest percentage of R+K (bin1; < 10% R+K), bin2 has 3033 proteins with moderate %R+K content (bin2; 10–13% R+K), and bin3 consists of the 1501 proteins, with very high %R+K (bin3; >13% R+K) (Fig 4A). We next asked if these bins were enriched for any groups of functional pathways (based on Gene Ontology). Bin1 and bin2 have disparate groups of GO terms, with no unique enrichment. However, bin3 was significantly enriched for ribosomal and translation related genes (Fig 4B). This arginine and lysine distribution in translation related genes are significantly higher than genome wide distributions (Wilcox test, p-value $< 10^{-10}$) (Fig 4C). Thus, translation related proteins are highly enriched for arginine and lysine amino acids.

Next, we asked if there is any correlation between the genes regulated by Gcn4 (in MM +Met), and the percentage of R+K encoded within these encoded proteins. Consistent with the repression of ribosomal genes by Gcn4, a significant proportion of the genes that are repressed by Gcn4 fall in bin3 (~40%, Fisher's exact test P-value $< 10^{-10}$) (Fig 4D and S5 Data). Therefore, a significant proportion of the genes induced by methionine, and further induced in Δ*gcn4* are arginine and lysine rich. This trend of a significant proportion of Gcn4 repressed genes being rich in arginine and lysine codon is consistent in starvation conditions as well (S13 Fig). This suggests the possibility of a more nuanced management of overall anabolism by Gcn4. The translation of arginine and lysine enriched proteins will be required for high translation, and this will require Gcn4-dependent precursors (arginine and lysine).

## Gcn4 dependent outputs can sustain high translation capacity during growth

Given this observation, we asked if, in a growth program, cells could sustain the synthesis of arginine and lysine rich genes if Gcn4 is absent. To evaluate this unambiguously, we designed inducible, luciferase-based reporters to estimate the translation of a several of R+K enriched genes, which are induced in cells by methionine (and further increased upon the loss of Gcn4) (S1 Data). We designed a plasmid, in which the gene of interest (GOI; amplified from the genomic DNA of *S. cerevisiae*) was cloned in frame with the luciferase coding sequence, in such a way that the entire fragment (GOI+Luciferase) will be under the control of an inducible promoter (S14 Fig). Using this system, measuring luciferase activity after induction will estimate the specific translation of the specific arginine/lysine enriched gene, in any condition.

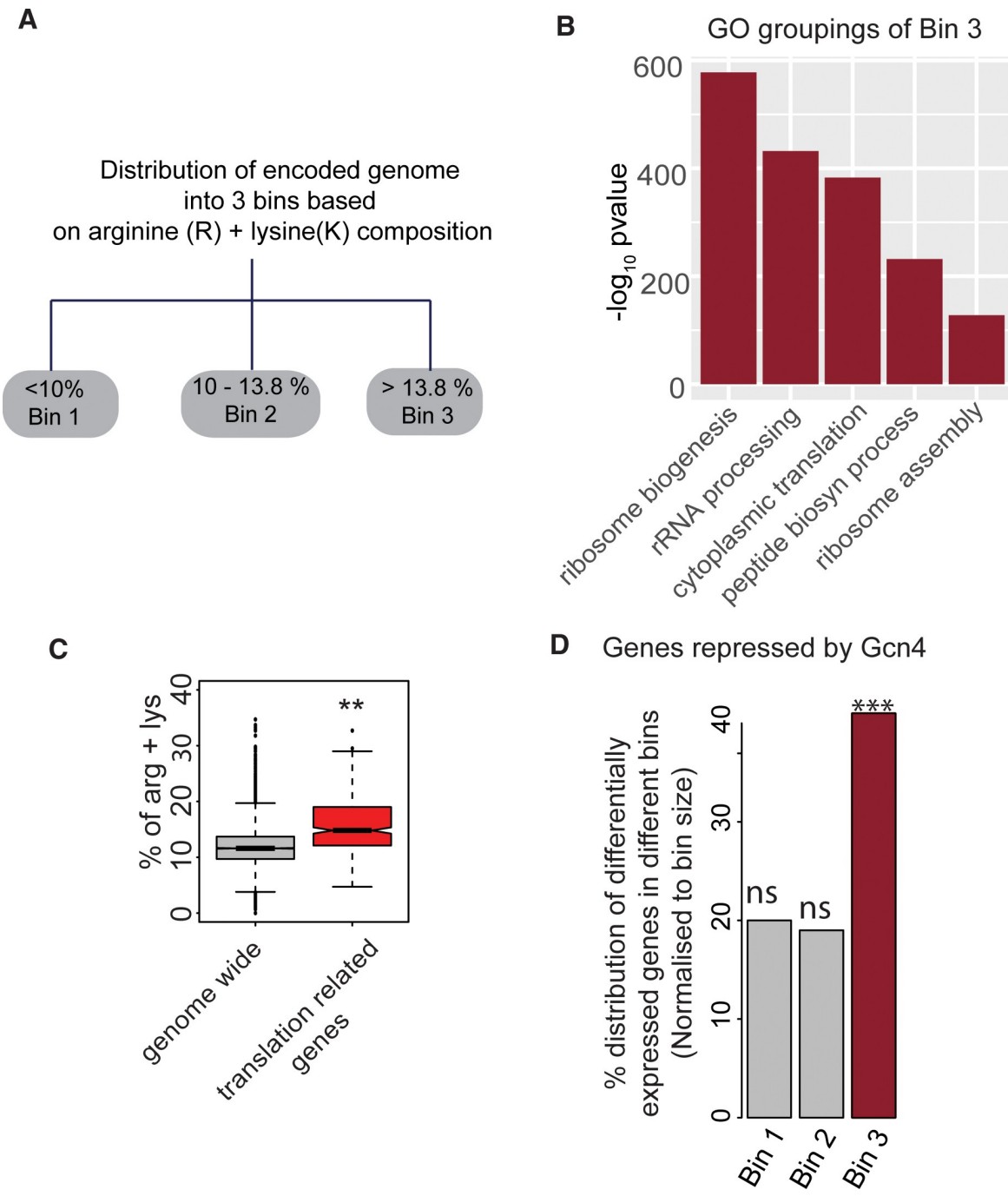

**Fig 4. Gcn4 globally represses arginine/lysine enriched genes, including the translational machinery.** A. "Binning" of the yeast proteome into three equal parts, based on the percentage of arginine and lysine in these proteins. The percentages of arginine and lysine (together) in these bins are indicated. B. GO based analysis reveals that bin3, which has the high percentage of arginine and lysine, is significantly enriched for ribosomal and translation related genes. The graph plots the most enriched GO term against -log$_{10}$(p-value). C. Boxplot, comparing the arginine and lysine composition of the entire proteome (excluding translation related genes), and the translation related genes. The translation related genes have a significantly higher than genome wide composition of arginine and lysine. D. Barplots, indicating in which bin (as shown in Fig 4A) the genes repressed by Gcn4 (i.e. induced in Δgcn4) fall under. A significant majority of the genes repressed by Gcn4 are enriched for arginine and lysine rich bin3 (Fisher exact test, p< 10$^{-10}$). Also see S13 Fig.

This accounts for only newly synthesized protein, and therefore avoids mis-interpretations coming from already existing protein in the cells before methionine addition. We made reporters for 4 such candidate genes (RPL32, STM1, NHP2, RPS20) (S14 Fig), all of which are strongly induced by methionine.

First, we contextualized the expression of these lysine and arginine enriched genes (based on reporter activity) in WT cells, under either a growth or a starvation regime where Gcn4 expression is high. The conditions we compared were MM (low Gcn4 expression), addition of methionine (growth program, strong Gcn4 induction), and the addition of 3-AT (amino acid starvation, high Gcn4). In these conditions, the reporters for Nhp2, Rpl32 and Rpl20 were induced for 30 min, and the luciferase activities were compared. Here, the luciferase activity of all three reporters significantly increases in methionine-supplemented conditions, and these decrease in the 3-AT condition (Fig 5A). This reiterates that the translational outcomes are entirely distinct in a growth or starvation program, despite high Gcn4 activity in both conditions.

We now could specifically determine the importance of Gcn4 activity upon the addition of methionine. First, we compared the extent of transcript expression for these lysine/arginine-enriched genes, Nhp2, Rpl32, Rpl20 and Stm1, in WT and Δ*gcn4* cells in the presence of methionine (Fig 5B). The loss of Gcn4 in these conditions further increased expression of these transcripts, reiterating the role of Gcn4 as a (indirect) repressor of these genes. We next directly measured the translation of these transcripts, using the luciferase-based reporters. For this, using a similar experimental setup as earlier, the luciferase activities in WT and Δ*gcn4* cells were measured after 30 minutes of induction with β-estradiol. All the reporters showed a 3–5 fold reduction in translation in *GCN4* deficient cells, compared to WT cells (Fig 5C). These data reveal that Gcn4 is critically required to maintain the translation capacity of the cell, during this growth program. Finally, to determine if this reduced translation capacity in Δ*gcn4* is due to the reduced supply of arginine and lysine in these conditions, we carried out rescue experiments with the addition of only these two amino acids. Both amino acids (2mM each) were provided to WT and Δ*gcn4* cells growing in the presence of methionine, and the luciferase activity (of the same reporters) was measured after induction. The supply of arginine and lysine significantly increased the translation of these reporter proteins (Fig 5D). To evaluate the specific requirement of arginine and lysine, we also included other amino acid controls. Here, three distinct amino acids (serine, glycine and alanine) which cannot easily interconvert to arginine or lysine, were supplemented as a pooled mixture along with methionine (all at 2mM each), and the luciferase reporter activity was measured. These amino acids *combined* had no effect on reporter activity in *Δgcn4* cells, and did not increase the translation of these reporter proteins (S15 Fig).

Collectively, we find that Gcn4 activity is central to sustain a growth program triggered by methionine. Gcn4 enables the supply of amino acids, particularly arginine and lysine, which are required for the translation of ribosomal proteins. Gcn4 also tunes the expression of these ribosomal transcripts. Together, this enables cells to maintain translational capacity, in order to sustain the anabolic program. This is in contrast to a starvation program due to amino acid limitation, where Gcn4 is also high. In such starvation contexts, the amounts of arginine and lysine enriched transcripts (including translation related transcripts) are low (and repressed by Gcn4), and the role of Gcn4 is primarily to restore amino acid levels.

## Discussion

A central theme emphasized in this study is mechanisms by which cells manage resource allocations, supply and demand during cell growth. Recent studies in organisms like yeast and *E.*

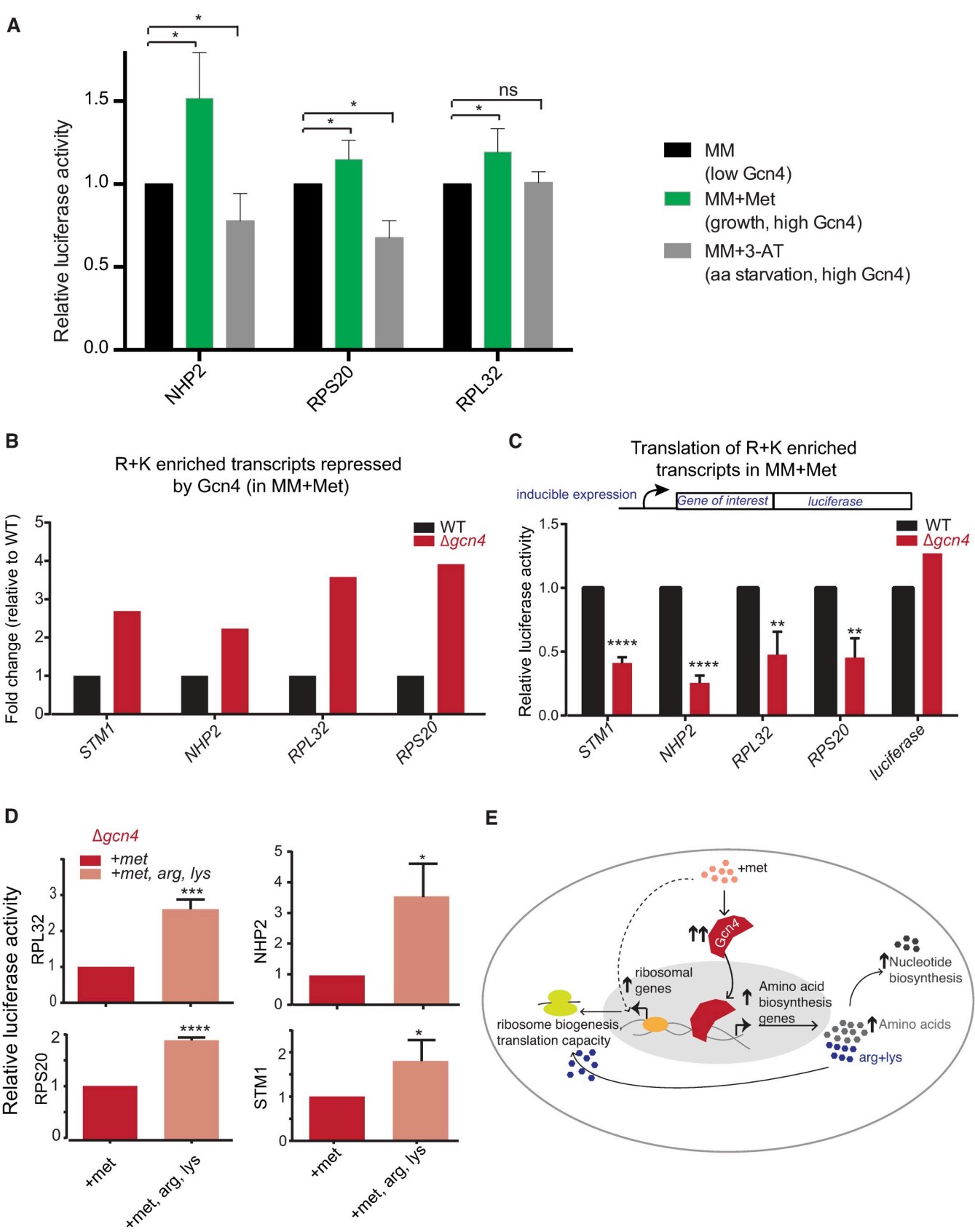

**Fig 5. Gcn4 dependent outputs can sustain high translation capacity during growth.** A. Translation of arg/lys genes increase during a growth program (methionine addition), and decrease during amino acid starvation (3-AT addition), both of which are conditions where Gcn4 is induced. Data show relative luciferase amounts, and are mean+/- SD from ≥ 3 biological replicates. $^*$p<0.05,$^{**}$p<0.01,$^{****}$<0.0001 (t-test). B. Arg/lys enriched genes are induced in *Δgcn4* cells in methionine supplemented medium. Barplots comparing relative transcript amounts for selected, highly induced, arginine and lysine enriched genes, between WT and *Δgcn4* cells. Data shown are taken from the RNA seq dataset. C. Arg/lys enriched gene transcripts cannot be translated in *Δgcn4* cells in methionine supplemented medium. Barplots, comparing the relative amount of proteins translated (in a methionine-dependent manner), for the genes shown in Fig 5B. These selected genes were cloned in frame with luciferase in an inducible system, to create a translation-reporter (S14 Fig), and translation of these were induced in WT or *Δgcn4* cells in methionine supplemented medium (MM+Met). Data show relative luciferase amounts, and are mean+/- SD from ≥ 3 biological replicates. $^*$p<0.05,$^{**}$p<0.01,$^{****}$<0.0001 (t-test). D. Supplementing arginine and lysine partly restores translational capacity in *Δgcn4* cells. Barplots, comparing the relative amount of proteins translated (in a methionine-dependent manner), for the genes shown in Fig 5C, in methionine-supplemented medium (MM+Met) or in MM+Met+Arg+lys. Data shown are mean+/- SD from ≥ 3 biological replicates. $^*$p<0.05,$^{***}$p<0.001,$^{****}$<0.0001 (t-test). Note, supplementing a combination of glycine, serine and alanine does not rescue luciferase reporter activity in *Δgcn4* cells (S15 Fig). E. A mechanistic model illustrating how high Gcn4 enables a methionine dependent anabolic response, by supplying amino acids, and maintaining translation capacity. Methionine induces Gcn4, and ribosomal transcripts. Gcn4 enables cells to temper ribosomal gene expression, and increases amino acid (particularly lysine and arginine) biosynthesis. This supply of amino acids (lysine and arginine) is critical to sustain the required translation of arginine/lysine enriched transcripts, which are significantly enriched for the translation machinery itself.

*coli* focus on protein translation, and the need to 'buffer' translation capacity during cell growth [8,51]. However, the process of cell growth requires not just translation reserves (in the form of ribosomes), but also a constant supply of biosynthetic precursors to meet high demand. This includes: amino acids to sustain translation as well as drive metabolic functions, nucleotide synthesis (for DNA replication, transcription and ribosome biogenesis), and sufficient reductive capacity (for reductive biosynthesis). While the production of ribosomes and translation are extremely resource-intensive processes [52], how the metabolic and biosynthetic supply of resources integrate with translation-related processes remain more poorly understood.

Here, using yeast, we obtain mechanistic insight into how Gcn4 enables cells to sustain the supply of biosynthetic precursors, during a growth program (induced by methionine). In this growth program, methionine induces genes involved in ribosomal biogenesis and translation [15], which are universal hallmarks of a growth program [22,52–55]. In addition, cells increase metabolic processes that sustain anabolism; primarily the pentose phosphate pathway, transamination reactions, an induction of amino acid and nucleotide synthesis [15]. In particular, through this study, we show how the Gcn4 transcription factor functions to critically support this growth program, by enabling high amino acid supply in order to maintain a sufficient translation capacity, as illustrated in a schematic model (Fig 5E). We can now define the roles of Gcn4 during a growth program, and contrast this with its roles in a starvation program. In a growth program (the methionine-induced context), the increased transcripts required for ribosome biogenesis and translation have a nuanced regulation by Gcn4. Gcn4 represses ribosomal genes (consistent with earlier reports), and in this context Gcn4 appears to moderate the extent to which ribosomal transcripts are induced. The methionine-dependent induction of ribosomal genes is likely controlled by directly activating the TOR pathway [13,56–58], which is a regulator of ribosomal biogenesis. However, despite the induction of ribosomal transcripts in *Δgcn4* cells (in this growth program), cells cannot sustain the required rates of protein synthesis, or maintain the high translation capacity required for growth. This is because the translation machinery itself is highly enriched for arginine and lysine amino acids. The restricted synthesis (and therefore supply) of these amino acids in *Δgcn4* cells will decrease the translation capacity in these mutants, despite an increase in ribosomal gene transcripts. In the presence of methionine, the increased synthesis (and therefore supply) of these two amino acids depends almost entirely on induced Gcn4. After all, to sustain high growth, and anabolic programs, cells need to maintain high rates of translation, and ribosomal capacity. Thus, through Gcn4, cells can couple translation with metabolism, and manage resource allocations to sustain anabolism.

Notably, the specific transcriptional role of Gcn4 in growth or starvation programs remains well conserved. Regardless of context, Gcn4 is required for amino acid biosynthesis (particularly lysine and arginine biosynthesis), and to repress ribosomal genes. However, the cellular outcomes of the Gcn4 dependent regulation of translation related genes are context dependent. In growth contexts (as induced by methionine), Gcn4 tempers the expression of already high ribosomal transcripts. During starvation, Gcn4 further represses ribosomal transcripts. As a result, the absolute abundance of the translation related transcripts are different under these two different growth states. This, when combined with the amounts of arginine and lysine (which are required to translate ribosomal transcripts), together account for the translation capacity observed in cells depending on the growth/starvation context. The context-dependent role of this transcriptional master-regulator therefore varies, across the spectrum of growth and starvation programs.

Multiple studies now support a role for Gcn4 during contexts of high growth, including recent reports on the required role of the mammalian ortholog of Gcn4 (ATF4) in driving cancers [31,32,59,60]. These studies suggest that ATF4 induction is critical for tumor progression during nutrient limitation, possibly by providing otherwise limiting metabolites [31,32]. Separately, observations over decades note that many rapidly proliferating tumors depend on methionine [61,62], and methionine restriction critically determines tumor progression [18,19]. In this study, we demonstrate how Gcn4 provides biosynthetic precursor supply to sustain anabolism, in an otherwise limiting environment. Speculatively, could the ability of methionine to induce proliferation in cancers rest upon the induction of ATF4, which controls the supply of amino acids and other biosynthetic precursors?

Summarizing, here we address Gcn4 function during a growth program triggered by methionine. This expands and elucidates the roles of this transcriptional master regulator in cells present in an anabolic state. This shows how, despite conserved function in both starvation and growth contexts, Gcn4 activity can lead to very different outcomes. Our study illustrates how cells manage the supply of important biosynthetic precursors, and regulate translation capacity and growth, when a specific metabolite cue induces high biosynthetic demands that need to be sustained.

## Materials and methods

### Strains and growth conditions

A fully prototrophic yeast strain *Saccharomyces cerevisiae* strain from a CEN.PK background [63] was used in all experiments. All strains used are listed in S1 Table, and plasmids used in S2 Table. For all the medium-shift experiments, overnight cultures were sub-cultured in YPD (1% Yeast extract and 2% Peptone, 2% Glucose) medium with an initial $OD_{600}$ of ~0.2. At an $OD_{600}$ of 0.8–0.9, cells were pelleted down (3500 rpm/90 seconds), washed and shifted to minimal medium MM (MM- Yeast Nitrogen Base with glucose as a carbon source) or MM+Met (MM with 2mM methionine). For the luciferase assay (described later), overnight cultures of cells in YPD with the antibiotic 1mM Nourseothricin (NAT) were used. The secondary culture was started with an initial $OD_{600}$ of ~0.5 in YPD + NAT and incubated at 30˚C and 250 rpm for 4 hours. Cells were then washed once in MM, and shifted to MM +Met, MM+Met+Arg +Lys or MM+Met+Gly+Ser+Ala. All amino acids were individually added at 2mM concentrations.

### Western blot analysis

Yeast cells with a 3x-FLAG epitope sequence chromosomally tagged at the carboxy-terminus of Gcn4 (endogenous locus) were used to quantify Gcn4 protein levels. Overnight cells were

sub-cultured in fresh YPD medium, with an initial OD of 0.2 and grown to an $OD_{600}$ of 0.8. Cells were pelleted down, pellets washed once in MM, re-harvested and shifted to MM and MM+Met after 1 hour of the shift. ~5 $OD_{600}$ of cells were harvested by centrifugation, and proteins were precipitated in 400 μl of 10% trichloro acetic acid (TCA), and extracted by bead beating with glass beads. Lysates were centrifuged to precipitate all proteins, and total protein pellets were resuspended in 400 μl of SDS-Glycerol sample buffer. Protein concentrations were quantified using Bicinconinic assay kit (G-Biosciences, 786–570) and equal concentrations of proteins were loaded into the 4–12% Bis-tris polyacrylamide gel (Thermo Fisher, NP0322BOX), resolved by electrophoresis, transferred to nitrocellulose membranes and detected by standard Western blotting using a monoclonal anti-FLAG M2- mouse primary antibody (Sigma Aldrich, F1804) and HRP labelled anti-mouse IgG secondary antibody (Cell Signaling technology, 7076S). Chemiluminescence reagent used: Advansta, K-12045. A different part of each gel was Coomassie stained in order to compare total protein loading amounts. Images were quantified using Image Quant.

## mRNA sequencing and data analysis

Overnight cultures of WT and *Δgcn4* cells were sub-cultured in YPD, with a starting $OD_{600}$ of 0.2 and grown to an $OD_{600}$ of 0.8–0.9. YPD grown cells were pelleted and washed once with MM. Washed cells were shifted to MM and MM+Met, and cells remained in this medium for ~1 hr. Cells were collected an hour after this shift and RNA was isolated by standard hot acid phenol methods. mRNA libraries were prepared using a TruSeq RNA library preparation kit V2 (Illumina). The quality of the libraries was analyzed using bioanalyser (Agilent 2100) and libraries were sequenced for 51 cycles using an Illumina HiSeq 2500 platform. For every experimental condition, data were obtained from two biological replicates. Normalized read counts between the biological replicates were well correlated (S1 Fig). For each strain we obtained ~30–35 million uniquely mapped reads. The raw data are available in NCBI-SRA under the accession PRJNA599001. The transcriptome data were aligned and mapped to the *S. cerevisiae S288C* genome downloaded from the saccharomyces genome database (SGD), using the Burrows-Wheeler Aligner [64] and the mapped reads with mapping quality of $\geq$ 20 were used for further analysis. The number of reads mapped to each gene was calculated and read count matrix was generated. The EdgeR package [65] was used for normalization and differential gene expression analysis. Differentially expressed genes with a fold change above 1.5 or 2 fold, with a stringent p-value cutoff of $< = 0.0001$ were considered for further analysis. Normalized read counts was calculated for every sample as described earlier [15]. Normalized read counts between the replicates are well correlated with the Pearson correlation coefficient (R) is more than 0.9 (S1 Fig). GO analysis of the differentially expressed genes were carried out using g: Profiler [66].

## Chromatin Immunoprecipitation sequencing and data analysis

a. Cell growth and sample collection: For ChIP sequencing, overnight cultures were re-inoculated in fresh YPD medium (RM), with an initial $OD_{600}$ of 0.2 and incubated at 30˚C until the $OD_{600}$ reached 0.8–0.9. Subsequently, 100 mL of cells were pelleted down, washed and shifted to MM and MM +Met. After 1 hour of the shift, cells were fixed using 1% formaldehyde, after which the fixing was quenched with 2.5M glycine.

b. Spheroplasting of fixed cells: Fixed cells were treated with 2-mercapto ethanol, and resuspended in 5 ml of spheroplasting buffer containing 1M sorbitol, 0.1M sodium citrate,

10mM EDTA, and distilled water, with 4mg/ml of lysing enzyme from *Trichoderma harzianum* (Sigma L1412-5G). This suspension was incubated at 37˚C for 5 hours.

c. Lysis and ChIP: Spheroplasts were pelleted down at 1000 rpm, washed 2X with Buffer 1 (0.25% Triton X100,10mM EDTA,0.5mM EGTA, 10mM sodium HEPES pH 6.5) and 2X with Buffer 2 (200mM NaCl, 1mM EDTA, 0.5mM EGTA, 10mM Sodium HEPES pH 6.5). Washed spheroplasts were resuspended in lysis buffer (50mM sodium HEPES pH 7.4, 1% Triton X, 140mM NaCl,0.1% Sodium deoxy cholate, 10mM EDTA) and lysis and DNA fragmentation were carried out using a bioruptor (Diagenode, Nextgen) for 30 cycles (30 sec on and off cycles). Lysates were centrifuged to remove the debris and clear supernatant was used for chromatin immunoprecipitation (ChIP). Immunoprecipitations were carried out by incubating the lysate with the monoclonal anti-FLAG M2- mouse primary antibody (Sigma Aldrich, F1804), and protein G Dynabead (Invitrogen, 10004D). Beads were washed sequentially in low salt, high salt and LiCl buffers, TE buffer and protein-DNA complex were eluted using elution buffer as reported earlier [67]. Decrosslinking of the immuno-precipitated proteins were carried out by using a high concentration of NaCl and incubating at 65˚C for 5 hours followed by proteinase-K treatment and DNA purification. Mock samples were prepared in parallel, excluding the antibody treatment. Libraries were prepared for the IP DNA and mock samples (NEBNext Ultra II DNA library preparation kit, Catalog no- E7103L) and sequenced using Illumina platform HiSeq 2500. Two biological replicates were maintained for all the samples. The raw data are available in NCBI-SRA under the accession ID PRJNA599001.

ChIP sequencing reads were mapped to the *S. cerevisiae S288C* genome downloaded from SGD. The reads with mapping quality < 20 were discarded, and the remaining reads were used for further analysis. The number of reads mapped to every 100bp non-overlapping bins were calculated using 'exomedepth' function of R-package GenomicRanges [68]. Read counts were normalized by dividing the number of reads falling within each bin by the total number of reads fall within the range of μ±x, where, μ = mode of the distribution of read counts of each bin, x = median absolute deviation of all the bins that has a number of reads that are less than the mean of the distribution. Subsequently, the regions with normalized read counts of above 2 were considered for further analysis. The binding regions separated by < 200bp were merged to give a single peak. Peaks which are conserved in both the replicates with the overlap of at least 50bp were considered as *bona fide* binding regions of Gcn4. Genes which are encoded around 750 bp on both sides of the peaks are listed in the S4 Data.

## Peak feature annotation and motif analysis

Genomic features of the peaks were identified using the annotatePeak.pl function of the HOMER tool [44]. For motif analysis, nucleotide sequences corresponding to the peak intervals were extracted from the genome and motif identification was performed using 'meme' function of MEME-suite [45].

## Direct and indirect target analysis

To annotate the genes corresponding to the peaks identified, the open reading frames encoded within 750 bp on both sides of the peak regions were considered as 'possible Gcn4 binding targets'. Gene sets which are differentially expressed in *Δgcn4* relative to WT under MM+Met condition, with a fold change of > 2 (~900 genes) were termed 'Gcn4 regulatory targets'. While comparing these gene lists, the genes which intersect between these two gene sets are

considered as 'direct Gcn4 binding targets' and the rest of the genes of 'Gcn4 regulatory targets' are 'indirect targets of Gcn4'.

## Metabolic flux analysis using LC/MS/MS

To determine if the rates of biosynthesis of amino acids are altered in MM+Met and Gcn4 dependent manner, we measured $^{15}$N-label incorporation in amino acids. We used $^{15}$N-ammonium sulfate with all nitrogen atoms labeled. Cells grown in YPD were shifted to fresh minimal medium (with the appropriate carbon source as indicated), containing 0.5 X of unlabelled ammonium sulfate (0.25%) and MM+Met containing 0.5X of unlabelled ammonium sulfate (0.25%). After 1 hour of shift to minimal media, cells were pulsed with 0.25% of $^{15}$N labelled ammonium sulfate and incubated for 15 minutes as indicated. After the $^{15}$N pulse, metabolites were extracted and label incorporation into amino acids was analyzed using targeted LC/MS/MS protocols as described earlier [15,69]. Extensive protocols and additional raw data for the MS experiment are provided in the S1 Text, S7 Data and S3 Table.

## Luciferase based translation reporters for lysine and arginine enriched genes

To measure the translation of specific transcripts that encode arginine and lysine enriched proteins, the ORF of the following proteins- RPL32, NHP2, STM1, RPS20- were amplified from genomic DNA isolated from WT *S. cerevis*iae. The amplified ORFs (without the stop codon) were ligated to the luciferase cDNA amplified from pGL3 (S2 Table). The resulting fragment with '$ORF_{RK\ rich\ genes}$ + *luciferase* were cloned in a centromeric (CEN.ARS) plasmid pSL207, a modified version of the plasmid used earlier [70]. Luciferase expression in this construct is under the control of a ß-estradiol inducible promoter [70]. The resulting plasmids with RPL32, NHP2, STM1, or RPS20 were cloned in frame with luciferase under the inducible GEV promoter, and named pSL218, pSL221, pSL224, pSL234 respectively (S14 Fig and S2 Table). SL217 (where only the luciferase cDNA amplified from pGL3 plasmid was cloned under the inducible promoter), was used as a control. These plasmids have ampicillin and Nourseothricin resistant cassettes (NAT$^r$) for selection. The plasmids were transformed to the WT and *Δgcn4* strains. To measure the translation of the genes cloned upstream of luciferase, strains carrying plasmids were grown in YPD overnight in the presence of NAT. Overnight grown cultures were shifted to fresh YPD+NAT with an initial OD$_{600}$ of 0.4, and grown for 4 hours at 30˚C. After 4 hrs of incubation in YPD, cells were washed and shifted to the MM and MM+Met. 75mM of 3-Amino triazole (3-AT) was used, wherever required. After 1 hour of the shift, cultures were split into two equal parts, one part of the culture was induced with 200nM ß-estradiol (Sigma Aldrich-E8875) and the other half was left as a mock-induced control. After 30 minutes of induction cells were harvested by centrifugation at 4˚C at 3000 rpm, washed 3X with lysis buffer (1X-PBS containing 1mM PMSF), and resuspended in 200µl of lysis buffer. Cells were lysed by bead beating at 4˚ C. After lysis, equal total protein concentrations of lysates were used while measuring the luciferase activity. Luciferase activity was measured using luciferase assay kit (Promega, E1500) and a luminometer (Sirius, Titertek Berthold Detection systems). Luciferase activities for each sample (Relative Light Units per Sec (RLU/sec)) were normalized with respective uninduced controls. Similar experiments were carried out under different media conditions supplemented with different amino acids, where the conditions are mentioned in the respective sections. The relative difference in luciferase activities between the strain types and media conditions were used to estimate changes in the active translation of these proteins.

## Statistical tests

R-Packages and Graph pad prism 7 were used for visualizing data and performing statistical tests. The EdgeR bioconductor packages was used for the differential expression analysis of RNA seq data, and GenomicRanges was used for ChIP seq and RNA seq analysis [68,71]. The respective statistical tests used, and the statistical significance is mentioned wherever required.

## Supporting information

**S1 Fig. RNA-seq correlation plots.** Correlation plots between replicates of the RNA sequencing data, from both WT and *Δgcn4* cells, for the indicated media conditions. Normalised read counts of the RNA sequencing data are plotted. Replicates shows good correlation with a Pearson correlation coefficient, R = 0.9.
(PDF)

**S2 Fig. Methionine induces a conserved transcriptional growth program irrespective of carbon sources.** A. Experimental design to study a growth program triggered by methionine. B. A volcano plot showing differentially expressed genes in MM+Met, relative to MM. Genes that are upregulated and downregulated in MM+Met (fold change of $\geq$ 1.5, p-value cut off of $10^{-4}$) are highlighted in red and blue respectively. C. A bar plot showing the most significantly enriched GO categories of the genes either induced or downregulated by methionine. The GO terms shown here are significantly enriched terms with the corrected p-value < 0.05 (hypergeometric test, Bonferroni Correction) (also see S3 Data for complete GO analysis results). For this, the numbers of genes induced (to the number of genes in that category) are also indicated within each bar. D. Heat maps showing the transcriptional induction of anabolic genes and translation related genes altered by methionine, i.e. in MM+Met relative to MM. Comparisons are made in two different carbon sources where methionine was added, glucose (this study) and lactate medium [5] (Also see S2 Data).
(PDF)

**S3 Fig. Quantitative LC-MS/MS based metabolic flux measurements to establish the methionine induced anabolic program.** (Left panel) Quantitative LC/MS/MS based targeted metabolic flux measurements, using $N^{15}$ ammonium sulfate labeling, to estimate nitrogen flux towards new amino acid synthesis in MM+Met compared to MM. The experiment was performed as illustrated in the schematic flow diagram. Data shown are the average of two biological replicates (with technical duplicates), with standard deviation. (Right panel) Quantitative LC/MS/MS based targeted metabolic flux measurements, using $C^{13}$ glucose labeling, to estimate carbon flux towards new nucleotide synthesis in MM+Met compared to MM. Data shown are the average of two biological replicates (with technical duplicates), with standard deviation. $^{*}$ p-value < 0.05, $^{**}$ p-value <0.01.
(PDF)

**S4 Fig. Serial dilution based growth assay, comparing WT and *Δgcn4* cells growing in the presence or absence of Methionine.** WT cells show better growth when supplemented with methionine. Deletion of Gcn4 lead to severely reduced growth in the presence of methionine. Also see Fig 1A.
(PDF)

**S5 Fig. Gcn4 protein expression, effect of loss of Gcn4 in MM and MM+Met, and Gcn4 ChIP-seq correlation plots.** A. A representative western blot for Gcn4-HA (Gcn4 tagged with the HA epitope at the endogenous locus) expression in the indicated medium. Gcn4 protein levels are high in MM+Met. Abbreviations: RM—rich medium, MM—minimal medium

without amino acids and with glucose as a carbon source, and MM+Met—minimal medium without amino acids and with glucose as a carbon source, supplemented with 2mM methionine. B. (left panel) A volcano plot comparing transcript expression in WT with *Δgcn4* cells (grown in MM+Met). Genes induced with a fold change cut-off of $\geq 2$, and p-value $\leq 10^{-4}$ are shown in red, and the downregulated transcripts (similar cut-off parameters) are shown in blue. The numbers of differentially expressed transcripts are also indicated. (right panel) A volcano plot comparing transcript expression in WT with *Δgcn4* cells (grown in MM. Genes induced with a fold change cut-off of $\geq 2$, and p-value $\leq 10^{-4}$ are shown in red, and the downregulated transcripts (similar cut-off parameters) are shown in blue. The numbers of differentially expressed transcripts are also shown. C. Correlation plots between normalised read counts of the ChIP sequencing data, from cells growing in different medium. Normalised read counts of 100000 randomly selected positions from the genome are plotted. ChIP data show enrichment/DNA binding for Gcn4 only in methionine supplemented conditions. Replicates show good correlation with a Pearson correlation coefficient, R = 0.9.
(PDF)

**S6 Fig. Venn diagrams showing the number of differentially expressed genes that overlap, from data obtained from distinct datasets of Gcn4 levels are high.** Genes differentially regulated by Gcn4 show significant overlaps between two distinct datasets, where Gcn4 was induced by different modes (Fisher exact test p $< 10^{-10}$, for both activated and repressed gene comparisons). The circles on the left use data from this study (methionine induced growth program), while the circles on the right use data from a study where, Gcn4 amounts were increased using an inducible system (2 hrs post induction of Gcn4) [7]. The overlapping genes between these two datasets were enriched for amino acids biosynthesis (activated in both the datasets) and the ribosomal protein (repressed in both the datasets). Also see S6 Data.
(PDF)

**S7 Fig. Read coverage for Gcn4 binding around the translation start sites of the target genes identified by ChIP seq, in a methionine dependent growth program.** A heatmap showing the read coverage from 1kb bases flanking the predicted translation start site (ATG) of genes, for the Gcn4 targets. The heatmap on the left shows read coverage in ChIP samples, and on the right shows coverage in mock-ChIP (control) in the MM +Met condition. The density plot in the top panel shows that most of the target genes have Gcn4 binding peaks upstream of the translation start site in ChIP samples (red) and such a pattern is absent in mock samples (blue). The deepTools package [8] was used for generating the matrix in this analysis.
(PDF)

**S8 Fig. ChIP qPCR validation of selected Gcn4 binding targets.** Gcn4 binding targets identified in the MM+Met condition were further confirmed by quantitative PCR, with the primers specific to the promoter region of the target genes. Gcn4 binding to the promoter of these targets is significantly high in MM+Met, relative to the MM. Data shown are mean±SD from 2 biological replicates. * p-value $< 0.05$.
(PDF)

**S9 Fig. Directly and indirectly regulated targets of Gcn4.** A. Pie chart showing the number of Gcn4 targets directly/indirectly repressed by Gcn4. B. List of genes directly repressed by Gcn4 and their reported function. C. Genomic tract showing the Gcn4 peaks for the selected genes that are activated directly by Gcn4 (top panel), and their transcript expression values as measured using RNA seq (bottom panel). D. Genomic tract showing the Gcn4 peaks for the selected genes that are directly repressed by Gcn4 (top panel) and their transcript expression

values as measured using RNA seq (bottom panel). The peaks were visualized using the Integrated Genome Browser (IGB) [9]. E. Integrated Genome Browser (IGB) tracts, showing Gcn4 binding peaks on the promoters of the genes that comprise the arginine, lysine and histidine biosynthetic pathways.
(PDF)

**S10 Fig. Plot showing correlations for transcript expression of either direct or indirect targets of Gcn4.** These are compared using datasets from two different conditions (x-axis this study, y axis [10]). The expression of direct targets of Gcn4 shows a Pearson correlation r = 0.43 between the datasets. The indirect targets show a Pearson correlation r = 0.3.
(PDF)

**S11 Fig. Correlation of ChIP binding signals obtained from distinct growth conditions.** For all the Gcn4 binding peaks identified in this study (x-axis), we compared the Gcn4 binding signal from two different growth conditions (y-axis left panel–from [10], right panel–from [11]) These data suggest that irrespective of the growth medium or condition, Gcn4 binding signals correlate between distinct datasets.
(PDF)

**S12 Fig. Gcn4 controls increased amino acid biosynthetic flux in the presence of methionine.** WT and *Δgcn4* cells grown in RM were shifted to MM or MM+Met with 0.5X of ammonium sulfate, and 0.5X of $N^{15}$ labeled ammonium sulfate, and metabolites extracted, 15 minutes after the pulse. Label incorporation into the indicated amino acids was measured using quantitative, targeted LC-MS/MS analysis (Detailed methods and raw data are provided in S1 Text and S7 Data).
(PDF)

**S13 Fig. Transcripts repressed by Gcn4 are enriched for arginine and lysine codons.** In the dataset of Gcn4 targets during amino acid starvation [10] the transcripts repressed by Gcn4 are also enriched for translation related processes, and these transcripts fall into the bin enriched for arginine and lysine codons. The left panel shows data from this study (as shown in Fig 4D), and the right panel shows a similar analysis using data from [10]. A significant enrichment of the Gcn4 targets (repressed by Gcn4) is present in bin3 (Fisher exact test: p-value $< 10^{-10}$).
(PDF)

**S14 Fig. Inducible luciferase-conjugated reporters to estimate expression of lysine and arginine enriched genes.** The complete open reading frames (ORFs) of the indicated genes were cloned in place of 'GOI-Gene of Interest' labelled in the plasmid map. In this study, ORFs of genes *RPL32*, *NHP2*, *STM1*, *RPS20* were cloned upstream of the luciferase sequence, and the resulting plasmids were named pSL218, pSL221, pSL224, pSL234 respectively. In this system, the expression of the GOI+luciferase is under the control of an inducible promoter, which is induced upon adding ß- estradiol, leading to gene expression and translation. The activity of luciferase enzyme (which is cloned in frame with the gene of interest (GOI)), serves as a quantitative indicator of the translation of that reporter in the given condition and genetic background. The plasmid map was created using SnapGene viewer version 3.3.4 (http://www.snapgene.com)
(PDF)

**S15 Fig. Supplementing non-arginine/lysine amino acids does not increase the translation of reporter genes in *Δgcn4* cells in MM+Met.** Related to Fig 5D: Cells were supplemented with serine, glycine and alanine in combination (as these amino acids do not interconvert to

arginine), in order to test if other amino acids could rescue the translation capacity of the *Δgcn4* in MM+Met. Glycine, serine and alanine (were supplemented at 2mM each), in similar experiments to those carried out with arg/lys, and reporter activity was measured. These pooled amino acids had no effect on reporter activity in *Δgcn4* cells.
(PDF)

**S1 Text. Supplementary results and methods.**
(DOCX)

**S1 Table. List of strains used in this study.**
(DOCX)

**S2 Table. List of plasmids used in this study.**
(DOCX)

**S3 Table. List of LC/MS/MS parameters (Q1/Q3) for metabolites detected.**
(DOCX)

**S1 Data. (.xlsx format), of differentially expressed genes in the indicated conditions.**
(XLSX)

**S2 Data. (.xlsx format), list of anabolic and translation related genes induced in the indicated conditions.**
(XLSX)

**S3 Data. (.xlsx format), of GO categories of genes up/down regulated in the indicated conditions.**
(XLSX)

**S4 Data. (.xlsx format), of Gcn4 targets based on ChIP-seq analysis.**
(XLSX)

**S5 Data. (.xlsx format), of genes repressed by Gcn4 that are in bin 3 (from Fig 4).**
(XLSX)

**S6 Data. (.xlsx format), of genes induced by Gcn4 and repressed by Gcn4, and overlap of Gcn4 targets from starvation studies.**
(XLSX)

**S7 Data. (.xlsx format) with mass spectrometry data for the $N^{15}$ label incorporation experiments measuring amino acid biosynthesis flux.**
(XLSX)

## Acknowledgments

We acknowledge extensive use of the NCBS/inStem/CCAMP next generation sequencing facilities, and the NCBS/inStem/CCAMP mass spectrometry facilities. We acknowledge Anjana Badrinarayanan and SL lab members for critical comments on the manuscript.

## Author Contributions

**Conceptualization:** Rajalakshmi Srinivasan, Sunil Laxman.

**Data curation:** Rajalakshmi Srinivasan, Adhish S. Walvekar, Zeenat Rashida, Sunil Laxman.

**Formal analysis:** Rajalakshmi Srinivasan, Adhish S. Walvekar, Zeenat Rashida, Aswin Seshasayee, Sunil Laxman.

**Funding acquisition:** Aswin Seshasayee, Sunil Laxman.

**Investigation:** Rajalakshmi Srinivasan.

**Methodology:** Rajalakshmi Srinivasan, Adhish S. Walvekar, Zeenat Rashida, Aswin Seshasayee, Sunil Laxman.

**Project administration:** Aswin Seshasayee, Sunil Laxman.

**Supervision:** Sunil Laxman.

**Validation:** Rajalakshmi Srinivasan.

**Visualization:** Rajalakshmi Srinivasan, Aswin Seshasayee, Sunil Laxman.

**Writing – original draft:** Rajalakshmi Srinivasan, Aswin Seshasayee, Sunil Laxman.

**Writing – review & editing:** Rajalakshmi Srinivasan, Aswin Seshasayee, Sunil Laxman.

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
