## [Decision Letter · Decision Letter 0]

24 Jul 2020

Dear Dr Laxman,

Thank you very much for submitting your Research Article entitled 'Genome-scale reconstruction of Gcn4/ATF4 networks driving a growth program' to PLOS Genetics. Your manuscript was fully evaluated at the editorial level and by three independent peer reviewers. The reviewers appreciated the attention to an important problem, but raised some substantial concerns about the current manuscript. Based on the reviews, we will not be able to accept this version of the manuscript, but we would be willing to review a much-revised version. We cannot, of course, promise publication at that time.

The reviewer comments can be found below.  As you can see Reviewer #1 is the most positive, Reviewer #3 is in the middle and Reviewer #2 is the most critical.  We as the you address all of their concerns, but here are some common/important threads from their comments:

1. Reviewer #2 notes that only 2 replicates were used for the metabolomics experiments - The Metabolomics Society recommends a higher number of replicates and suggests that 3 is the bare minimum that should be considered for publication.  In order to ensure rigor and reproducibility we will require this critique be addressed with additional experimentation.

2. A common theme in the reviewer comments is the need to tone-down the text.  Even the most complimentary reviewers/editors felt the use of exaggerative language was hurting rather than helping the manuscript.

3. Some of the reviewers noted English usage errors - please note that should your manuscript be accepted PLOS does not offer proof reading services so it is critical that these issues be corrected during the revision process.  Some authors find it helpful to have the manuscript proof read by colleagues while others uses professional editing services.  Similarly one reviewer noted that the references needed serious editing - please be sure this is addressed.

4. Two of the reviewers commented on data availability.  Please note that PLOS Genetics requires all supporting data to either be presented in the paper, in the supplementary material, or deposited in a stable public database.  This requirement is non-negotiable and must be addressed.

If you decide to revise the manuscript for further consideration at PLOS Genetics, please aim to resubmit within the next 60 days, unless it will take extra time to address the concerns of the reviewers, in which case we would appreciate an expected resubmission date by email to plosgenetics@plos.org.

[LINK]

We are sorry that we cannot be more positive about your manuscript at this stage. Please do not hesitate to contact us if you have any concerns or questions.

Yours sincerely,

Gregory P. Copenhaver

Editor-in-Chief

PLOS Genetics

Reviewer's Responses to Questions

**Comments to the Authors:**

Reviewer #1: I liked this paper quite a bit, so my comments are minor. It's well-written, and the analyses are appropriate. The ChIP data in Figure 2 are striking and convincing. Given Gcn4's central role in amino acid biosynthesis, there is at least one control I'd like the authors to carry out to establish the privileged nature of lys/arg being the limiting building blocks in increasing protein production in gcn4∆ cells.

Comments:

Figure 5D: The authors need to test if other amino acid supplements beyond arg/lys increase reporter output (it would also be good to test arg and lys separately). If they are truly privileged, it might even be worth adding this result to the title of the paper.

Page 3, Line 111: Is methionine the only amino acid that induces a "growth signature" expression program?

Page 3, Line 127: Can the overlap between the response on lactate/glucose be further quantified? "overlapped well" is a little too qualitative.

A recent publication from Hackett et al includes gene expression profiling of cells following induction of GCN4 (https://www.embopress.org/doi/10.15252/msb.20199174) -- it would be interesting to see if if there is a signature of ribosomal gene repression in those data. These data have a benefit that they induce a single TF without other chemical perturbations, so the direct/immediate effects of GCN4 activation can be gleaned. It would also be interesting to compare the GCN4 induction data to your ChIP results.

Figure 1 legend mentions binding "Gcn4 binds to its target gene promoters related to metabolism during a growth program", but the figure itself is about expression, not binding.

Reviewer #2: The manuscript (MS ID: PGENETICS-D-20-00597) entitled 'Genome-scale reconstruction of Gcn4/ATF4 networks driving a growth program', which has been submitted to PLOS Genetics by Srinivisan et al., illustrate how growth and starvation outcomes are controlled using the same Gcn4 transcriptional outputs. Relevant and timely study. However, the following comments / suggestions/ advice are provided to help the authors improve on the currently submitted version of the manuscript only.

Critique and concerns for improvement:

A lot is left for imagination : “using a targeted LC/MS/MS based approach [56] to measure amino acid synthesis flux, based on stable-isotope incorporation.”

Metabolomics raw data can be shared/ provided in popular data repositories dedicated for metabolomics data.

L666: Which R-packages were used? Name them for individual tasks. Also, for FAIR practices provide codes as supplementary materials.

L671: Remove the word “extensive”…

L630: instead of citing references: [20,56] provide details as supplementary materials or leave the protocols as Protocols.io and then cite them in this manuscript.

Major concern: L550: “platform. For every experimental condition, data were obtained from two biological replicates.” N=2 is not standard and reproducible. How would the authors defend their work in terms of effect size and statistical robustness/ considerations ?

Discussion is too short.

Results sections are “too long” instead laced with too much citations, discussions and interpretations.

Materials and methods are very well detailed.

Unsure why 2 sets of references have been added to the document and needs to be amended.

Figure 1B: Use “transcripts” instead of genes.

Major language related improvement needed. Present version has some common, punctuation, tense, grammatical, typological sentence framing/ phrase construction issues which should be taken care by the authors/ collaborators. Many loosely constructed sentences floating around.

Please change and improve upon the conclusion to provide very specific and key take-home messages to the readers and should be only based on the findings and not extrapolations/ future aspects. Currently too long.

References are NOT OK and poorly organized in terms of formats. Follow the journal format typically as instructed to the authors. Check all the references one by one manually, so that everything cited in the text are also listed in the list of references.

If at all a revised version is submitted, then please make all changes in differently colored fonts (red/blue) or highlighted background (yellow) so that it would save the reviewer's time in finding the changes with lesser efforts and time.

Please also incorporate all the answers/ discussions/ points raised by this reviewer into the manuscript rather than just explaining to this reviewer- in order to make the manuscript stronger than it is now.

Reviewer #3: The authors elegantly characterise the role of GCN4 in growth conditions. They show that it indeed regulates a similar set of genes as during starvation (for which the gene is much better known). They propose an interesting indirect regulatory mechanism whereby GCN4 regulates the production of specific amino acids (arginine and lysine), which are specifically enriched in a set of genes translation related genes. They thus propose that GCN4, through regulation of its direct targets, also indirectly regulates the translation of translation genes.

Overall, the study is well carried out, follows a logic flow and reveals a conceptually interesting regulatory mechanism. Throughout the manuscript - in particularly in the beginning - there are several overinterpretations of their results, so the authors should go over the text again and make sure the conclusions they draw at each stage are indeed supported by the data they present. The study is very nice and the results very solid, thus making those overstatements just hurts the credibility of the latter parts.

Specific comments:

- please remove all the "striking", "strikingly" etc. There is also no need to keep repeating that the starvation program has been studied before i.e. all the "conventional" in front of starvation program can be removed, it suffices to state the in the intro and then just use it as comparison.

- the beginning of the study feels like the authors are building a strawman (GCN4 has only been studied in stress, so we are doing something else). I would find it conceptually more attractive to build the story about context-dependency of GCN4 (which is essentially what the authors study). That is of course only a suggestion.

- Fig 1: instead of the simple overlaps in the venn diagram it would be more informative to show how the fold-changes correlate. Is it in principle the same gene expression program that gets activated? That is very difficult to judge from the Venn diagram.

- similarly for the direct vs indirect targets, do the direct target genes (i.e. those with a GCN4 motif or ChIP-seq peak) correlate better between starvation and growth, than the indirect ones?

- Fig 2B: what are the genes displayed, are these only those that are differentially expressed in GCN4? Should be clarified in the legend. How is the heatmap sorted?

- Fig 2E: what is the interpretation of the differences between ORF and Non-coding across the different studies? Also the axes on the figures are very different (0-35 for this study vs 0-500 and 0-300) for previous studies. What is the relevance of this? Overall it is not clear what the message of this figure is. Despite the use of "strikingly" in the associated sentence it is not clear to me what this result means.

- Fig 3A: how does this look for the starvation program? Are the direct genes the same? This is important for interpreting the context-dependent role of GCN4.

- Fig 3E/F, similar comment to above - instead of simple overlaps the authors should show scatterplots of fold-changes to allow readers to interpret the full data.

- Fig 4: similar to above - show the starvation response GCN4 targets a similar enrichment? I.e. is this mechanism conserved across different cellular context?

Some examples of conclusions that are not well supported by the data presented at the stage of the manuscript:

- "This strikingly revealed that the transcripts associated with protein translation, which were already induced by methionine, further increase in the absence of Gcn4. i.e. Gcn4 partially represses cytoplasmic translation even in a growth program. " is based on GO term analysis. The authors should who what the actual genes are and then draw a conclusion (GO terms are not very exact in saying what a set of genes is actually doing, since it is always an enrichment)

- "Collectively, this reveals that Gcn4 is essential for the induction of genes involved in these metabolic processes, which is a majority of the methionine induced anabolic program, but partially represses translation." This is a complete overstatement. Where does the evidence for "essential" come from? We are talking about an up/down regulation, not a switch on/off.

**Have all data underlying the figures and results presented in the manuscript been provided?**

Reviewer #1: Yes

Reviewer #2: Yes

Reviewer #3: **No: **Raw ChIP-seq data seems not deposited

PLOS authors have the option to publish the peer review history of their article (what does this mean?). If published, this will include your full peer review and any attached files.

Reviewer #1: No

Reviewer #2: No

Reviewer #3: No

---

## [Decision Letter · Decision Letter 1]

4 Nov 2020

Dear Dr Laxman,

We are pleased to inform you that your manuscript entitled "Genome-scale reconstruction of Gcn4/ATF4 networks driving a growth program" has been editorially accepted for publication in PLOS Genetics. Congratulations!

Yours sincerely,

Gregory P. Copenhaver, Ph.D.

Editor-in-Chief

PLOS Genetics

Gregory Copenhaver

Editor-in-Chief

PLOS Genetics

Comments from the reviewers (if applicable):

Reviewer's Responses to Questions

**Comments to the Authors:**

Reviewer #1: The authors addressed the reviewers concerns, but I believe the new genomics data have not been uploaded to a public database (like GEO). Once the fastq's and processed versions of the data are fully public, I recommend acceptance.

Reviewer #2: None.

**Have all data underlying the figures and results presented in the manuscript been provided?**

Reviewer #1: **No: **ChIP-seq / RNA-seq data need to be put in public database

Reviewer #2: Yes

PLOS authors have the option to publish the peer review history of their article (what does this mean?). If published, this will include your full peer review and any attached files.

Reviewer #1: No

Reviewer #2: No

**Data Deposition**

http://datadryad.org/submit?journalID=pgenetics&manu=PGENETICS-D-20-00597R1

**Press Queries**

---

## [Editor Report · Acceptance letter]

9 Dec 2020

PGENETICS-D-20-00597R1 

Genome-scale reconstruction of Gcn4/ATF4 networks driving a growth program 

Dear Dr Laxman, 

We are pleased to inform you that your manuscript entitled "Genome-scale reconstruction of Gcn4/ATF4 networks driving a growth program" has been formally accepted for publication in PLOS Genetics! Your manuscript is now with our production department and you will be notified of the publication date in due course.

With kind regards,

Nicola Davies

PLOS Genetics

On behalf of:
